# ARMOR: Conceptual Augmentation for Robust Multi-Concept Erasure in Stable Diffusion via Model Retrieval

## Abstract

Stable Diffusion enables high-quality synthesis but raises risks around copyright, misinformation, and explicit content. Concept erasure helps mitigate these risks by fine-tuning model weights, yet existing methods face two key challenges: (1) **Robustness**: erased concepts can be reconstructed via synonymous representations or adversarial attacks, and (2) **Multi-concept erasure**: training a single model to erase multiple concepts often strongly perturbs the weights, leading to degraded general-purpose generation. To address these challenges, we introduce **ARMOR**, a novel framework that integrates **conceptual augmentation** with a **model retrieval** approach for robust multi-concept erasure. Our method introduces two key innovations: First, we propose a conceptual augmentation technique that distils visual concepts into text modality for more effective and robust fine-tuning. Second, for each concept, we fine-tune the cross-attention key/value projection layers to obtain a dedicated eraser, and employ a retrieval mechanism that dynamically selects the appropriate erasers at inference, achieving a superior removal–generation trade-off. Extensive experimental results demonstrate that ARMOR outperforms prior work on challenging multi-concept erase tasks, resists red-team attacks, and achieves the best CLIPScore gaps, with **at least 10% gains over the second best** across four tasks.

## 1 Introduction

Text-to-image models (Rombach et al., 2022; Zhang et al., 2023; Zhao et al., 2024a;b) have shown impressive capabilities in generating realistic and diverse images. However, they also introduce significant challenges, such as the infringement of copyright, the creation of fake news, and the generation of explicit content, all of which raise important ethical and societal concerns (Abdikhakimov, 2023; He et al., 2024; Schramowski et al., 2023; Jiang et al., 2023). These issues highlight the need for responsible deployment and regulation of such advanced technologies.

Researchers have proposed several countermeasures, such as dataset cleaning (Carlini et al., 2022; Rombach, 2022; Rombach & Esser, 2022), post-filtering (OpenAI, 2023; Rando et al., 2022) and concept erasure (Heng & Soh, 2023; Kim et al., 2023; Kumari et al., 2023; Zhang et al., 2024a) to address these challenges. Among these methods, concept erasure prevents models from generating undesired content by fine-tuning model weights to replace unwanted concepts (*e.g.,* Van Gogh's art style) with surrogate concepts (*e.g.,* cartoon style). Compared to dataset cleaning and post-filtering, concept erasure is more effective and harder to bypass, making it a promising approach for restricting harmful or sensitive content generation. However, existing concept erasure methods still face two major challenges. First, *robustness* is a key issue. Erased concepts can often be recovered through euphemistic phrasing or adversarial prompts (Zhang et al., 2025; Pham et al., 2023; Feffer et al., 2024; Ma et al., 2024), reducing the erasure's effectiveness. Second, *multi-concept erasure* remains challenging. Fine-tuning in current methods can impair general-purpose generation, and this degradation becomes more pronounced as more concepts are erased , leading to catastrophic forgetting (Shin et al., 2017; Kirkpatrick et al., 2017).

To address these two key challenges, we propose the **ARMOR**, leveraging Conceptual **A**ugmentation for **R**obust **M**ulti-Concept Erasure in Stable Diffusion via **Mo**del **R**etrieval.

Most concept erasure methods (Gandikota et al., 2023; 2024; Lyu et al., 2024; Zhao et al., 2024c) only rely on human-interpretable textual prompts, which inherently provide only a narrow coverage of the textual space. However, diffusion models can also respond to semantically opaque or even unintelligible token sequences, meaning that human-readable text captures only a limited subset of the underlying concept space (see Fig. 1, left). This restricted textual representation reduces robustness against adversarial attacks. To address this limitation, we propose a concept augmentation strategy that leverages concept-related images to explore text embeddings beyond human semantics. Concretely, we back-optimize selected text tokens against concept images to derive augmented text embeddings that encode image-grounded features—including tokens unintelligible to humans but effective for

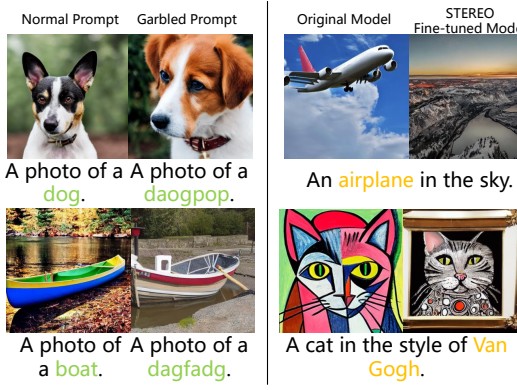

Figure 1: Limits of human-interpretable text (left) and degradation from naive fine-tuning (right). Our method augments text with image-derived tokens for robustness and employs retrieval to balance erasure with generation.

the model. Joint training on original and augmented texts expands semantic coverage, improving both generalization and robustness. For multi-concept erasure, a straightforward strategy is to erase multiple concepts simultaneously within a single model, yet this often compromises generative capacity (see Fig. 1, right), leading to catastrophic forgetting. To address these limitations, we fine-tune a dedicated set of cross-attention key–value parameters in the U-Net for each concept, and introduce a contrastive learning–based retrieval module that routes prompts to the most appropriate models during inference. This modular design minimizes interference across concepts and preserves strong performance even under large-scale concept removal.

Experimental results show that our method excels across four erasure tasks: object, explicit content, celebrity, and artistic style erasure. Furthermore, it demonstrates high effectiveness even when confronted with red-team attack methods (Zhang et al., 2025; Tsai et al., 2024; Pham et al., 2023) and when erasing over a hundred concepts. Our main contributions are summarized as follows:

- **Conceptual Augmentation for Robust Fine-Tuning**: We propose a conceptual augmentation technique that condenses visual concepts into a text-based modality, enhancing the robustness of model fine-tuning for effective concept erasure.

- **Model Retrieval for Precise Erasure**: We develop dedicated erasure models for each undesired concept and a retrieval module that dynamically selects the appropriate models during inference, optimizing the balance between removing unwanted concepts and preserving generative capacity.

- **State-of-the-Art Performance in Multi-Concept Erasure**: Extensive experiments show that ARMOR outperforms existing methods on challenging multi-concept erasure tasks, including object erasure, explicit content erasure, celebrity erasure, and artistic style erasure.

## 2 RELATED WORK

**Concept Erasure in Stable Diffusion.** Among existing security measures for Stable Diffusion, dataset filtering and image post-processing are either costly or easily bypassed. In contrast, concept erasure, which fine-tunes model weights to suppress undesired content by aligning target concepts with blank or surrogate texts, has shown promise. However, these methods are vulnerable to red-team attacks and prone to catastrophic forgetting when erasing many concepts. Although some works explore multi-concept erasure (Gandikota et al., 2024; Lu et al., 2024; Lyu et al., 2024; Zhao et al., 2024c), such methods often falter when scaling to hundreds of concepts. Other approaches improve robustness via model pruning or adversarial learning (Zhang et al., 2024b; Srivatsan et al., 2024; Gong et al., 2025; Kim et al., 2024; Huang et al., 2024), such as in (Chavhan et al., 2024), which deactivates parameters closely tied to unwanted concepts during inference. Yet, these robustness methods do not support multi-concept erasure. Our approach addresses these gaps by leveraging conceptual augmentation for enhanced robustness and employing dedicated fine-tuning with model retrieval to support a large number of concepts to be erased.

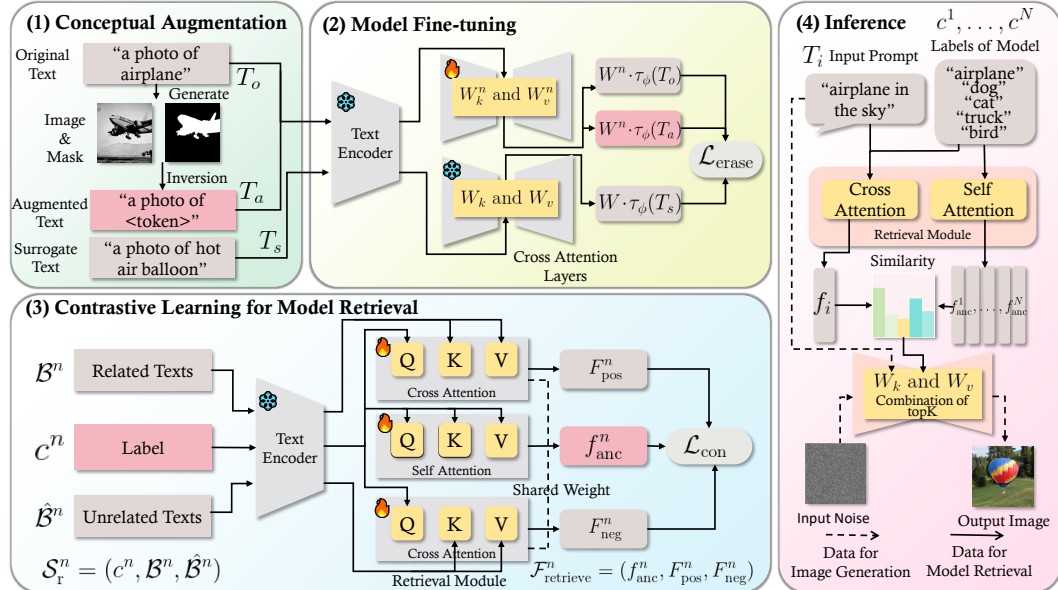

Figure 2: **Overview of the proposed ARMOR framework. (1) & (2) Conceptual Augmentation for Robust Fine-Tuning**: Apply a closed-form update to the U-Net cross-attention key/value projection layers to align the target (to-be-erased) concept with a surrogate, while optimizing image-based tokens to improve robustness. **(3) Contrastive Learning for Model Retrieval**: Optimizing the retrieval module through contrastive learning, where prompts for erased concepts, unrelated concepts, and concept labels are mapped into positive samples, negative samples, and anchors, respectively. **(4) Inference**: At inference, given the input prompt, the retrieval module selects the suitable sets of cross-attention key/value projection parameter pairs to prevent the generation of undesired concepts.

**Attacks against Stable Diffusion.** To induce Stable Diffusion to generate insecure content, various methods have been devised to target the model's security mechanisms and craft offensive prompts Pham et al. (2023); Zhang et al. (2025); Tsai et al. (2024). For example, UnDiff Zhang et al. (2025) is a typical white-box attack that leverages access to the model's internal weights—treating the diffusion model as a classifier—and uses gradient-based optimization to generate prompts most likely to yield insecure outputs. However, in practical settings, attackers often lack access to these internal weights, leading to the emergence of black-box attack methods. In this context, Ring-A-Bell Tsai et al. (2024) employs CLIP to extract feature representations associated with insecure content without accessing the model's weights, subsequently fusing these features with benign text to construct offensive prompts. Such attack strategies introduce novel security challenges for Stable Diffusion and underscore the pressing need for robust protection measures.

## 3 METHODOLOGY

### 3.1 MOTIVATION

It is observed that while existing concept erasure methods are effective in reducing the generation of relevant concepts, they remain underdeveloped in the following two points:

• **Robustness**: Once a model erases specific concepts, these concepts are frequently recoverable through synonymous representations or targeted red-team attack strategies. Such recoverability undermines the model's ability to effectively and permanently eliminate undesired concepts, posing a significant weakness in its robustness.

• **Multi-concept erasure**: When mass concepts are erased, the erasure process often proves ineffective. In many cases, the model fails to remove these concepts, or it suffers from catastrophic forgetting, where previously learned information is unintentionally lost. This issue highlights the difficulty in maintaining model stability while performing large-scale concept erasure.

To address the two challenges above, we propose ARMOR, which achieves robust erasure of mass concepts utilizing conceptual augmentation and model retrieval.

## 3.2 OVERVIEW

Our goal is to erase the model's ability to generate multiple concepts by fine-tuning only the text-related cross-attention modules $W_k$ and $W_v$ in a diffusion model $\epsilon_\theta$ with a text encoder $\tau_\phi$. The method consists of four stages (see Fig. 2).

**Erase dataset.** For each concept $c^n$, we generate $K$ images from an original prompt $T_o$, and derive an augmented text $T_a^k$ from each image. These augmented texts form the set $T_a$, where $T_a^k$ denotes the $k$-th element( Throughout this paper, we use uppercase letters to denote sets, and superscripts $k$ or $n$ to index specific elements.). Each $T_a^k$ is paired with the surrogate prompt $T_s$ and the original prompt $T_o$ to create erasure triplets, which together constitute the erasure set $\mathcal{S}_\mathrm{e}$:

$$\mathcal{S}_\mathrm{e} = \{(T_o, T_a, T_s) \mid 1 \le k \le K\}. \tag{1}$$

**Per-concept fine-tuning.** For each concept's dataset $\mathcal{S}_\mathrm{e}^n$ and a preservation dataset $\mathcal{S}_\mathrm{p}$ of $\tilde{K}$ retained concepts, we fine-tune $(W_k, W_v)$ by minimizing the erasure loss:

$$\mathcal{W} = \left\{ (W_k^n, W_v^n) \mid 1 \le n \le N, (W_k^n, W_v^n) = \arg\min_{(W_k, W_v)} \mathcal{L}_\mathrm{erase}(\mathcal{S}_\mathrm{e}^n, \mathcal{S}_\mathrm{p}, W_k, W_v) \right\}. \tag{2}$$

**Contrastive learning for model retrieval.** To retrieve the most suitable models from $\mathcal{W}$ during inference, we construct a set of contrastive training subsets $\mathcal{S}_\mathrm{r} = \{\mathcal{S}_\mathrm{r}^1, \mathcal{S}_\mathrm{r}^2, \ldots, \mathcal{S}_\mathrm{r}^N\}$, one for each concept $c^n$. Each subset $\mathcal{S}_\mathrm{r}^n = (c^n, \mathcal{B}^n, \hat{\mathcal{B}}^n)$ consists of a concept label $c^n$, a set of related texts $\mathcal{B}^n = \{P^k\}$ where each $P^k$ is associated with $c^n$, and a set of unrelated texts $\hat{\mathcal{B}}^n = \{\hat{P}^k\}$. Using cross-attention $\mathcal{A}_c$ for samples and self-attention $\mathcal{A}_s$ for anchors, we extract features: $\mathcal{F}_\mathrm{retrieve}^n = (f_\mathrm{anc}^n, F_\mathrm{pos}^n, F_\mathrm{neg}^n)$ from $\mathcal{S}_\mathrm{r}^n$ and optimize the contrastive loss:

$$\mathcal{A}_c, \mathcal{A}_s = \arg\min_{(\mathcal{A}_c, \mathcal{A}_s)} \mathcal{L}_\mathrm{con}(\mathcal{A}_c, \mathcal{A}_s, \mathcal{S}_\mathrm{r}). \tag{3}$$

**Test-time model retrieval.** During inference, for each model $(W_k^n, W_v^n)$ with label $c^n$, we compute similarity between the input prompt feature $f_i^n = \mathrm{CAtten}(T_i, c^n)$ and the anchor feature $f_\mathrm{anc}^n = \mathrm{SAtten}(c^n)$. The models with the highest similarity are selected:

$$(W_k^*, W_v^*) = (W_k, W_v) + \alpha \sum_{n \in \mathrm{TopK}_n[\mathrm{Sim}(f_i^n, f_\mathrm{anc}^n)]} ((W_k^n, W_v^n) - (W_k, W_v)). \tag{4}$$

The final model $(W_k^*, W_v^*)$ can replace unwanted concepts in prompt $T_i$ with surrogate ones during generation, ensuring effective concept erasure. Notably, our retrieval-based approach not only enables compositional erasure of multiple concepts but also minimizes the impact on model weights when generating general-purpose concepts. Further details follow in the subsequent sections.

## 3.3 CONCEPT-AUGMENTED DATASET CONSTRUCTION

Most concept erasure methods (Gandikota et al., 2023; 2024; Kumari et al., 2023; Heng & Soh, 2023; Lyu et al., 2024) typically depend on text prompts that explicitly specify the target concept, assuming concepts can be fully captured by language. However, many concepts span broader semantic spaces that simple text alone cannot represent. To address this, we propose conceptual augmentation, which leverages visual cues for richer and more comprehensive concept representations.

Given a concept, we first generate $K$ relevant images based on the original text prompts $T_o$, capturing various visual aspects of the concept. We then apply textual inversion (Gal et al., 2022) to distill meaningful visual information into tokens. By optimizing these tokens, we obtain augmented text prompts $T_a^k$ for each $k$-th image, enhancing the expressive capacity of $T_o$ and providing a richer concept representation. We refine the extracted information by masking each image, which reduces irrelevant background details. The process is formulated as:

$$\min_{T_a^k} \mathbb{E}_{z_t \sim \mathcal{E}(x^k), t, T_o, \epsilon \sim \mathcal{N}(0,1)} \left[ \|\mathrm{Atten}(z_t, \tau_\phi(T_a^k)) - M^k\|_2^2 + \|\epsilon M^k - \epsilon_\theta(z_t, t, \tau_\phi(T_a^k)M^k\|_2^2 \right], \forall 1 \le k \le K, \tag{5}$$

where $z_t$ is the latent feature of the input image $x^k$ at timestep $t$, $\epsilon$ is the noise sampled from a normal distribution $\mathcal{N}(0, 1)$, $\mathrm{Atten}(z_t, \tau_\phi(T_a))$ denotes the attention score of cross-attention layers in U-Net, $\tau_\phi$ is the text encoder, $M^k$ is the mask added to each $k$-th image.

As described in Eq. 1, the original prompt $T_o$ and augmented prompts $T_a = \{T_a^k\}_{k=1}^K$ are combined with a surrogate concept $T_s$ to construct the erasure dataset $\mathcal{S}_\mathrm{e}$, which will be used in subsequent sections.

### 3.4 Per-concept Model Fine-tuning

In text-to-image generation, the input text serves as a conditioning signal primarily affecting the cross-attention layers of the U-Net. Here, image features act as queries, while text embeddings provide keys and values. Since the cross-attention map matrices $W_k$ and $W_v$ directly govern this interaction, we fine-tune only these parameters while freezing the rest of the model. This approach effectively aligns undesired concepts with their surrogates while minimizing the computational cost of fine-tuning. When applied to the augmented erasure dataset $\mathcal{S}_e$ and the preservation dataset $\mathcal{S}_p$, the loss function to be minimized during fine-tuning is defined as follows:

$$\mathcal{L}_{\text{erase}}(\mathcal{S}_e, \mathcal{S}_p, W_k, W_v) = \mathcal{L}(\mathcal{S}_e, \mathcal{S}_p, W_k^n, W_k) + \mathcal{L}(\mathcal{S}_e, \mathcal{S}_p, W_v^n, W_v). \tag{6}$$

The formulas for key and value have the same form and are expressed as follows:

$$\mathcal{L}(\mathcal{S}_e, \mathcal{S}_p, W^n, W) = \underbrace{\|W^n \cdot \tau_\phi(T_o) - W \cdot \tau_\phi(T_s)\|_2^2 + \sum_{k=1}^{K} \|W^n \cdot \tau_\phi(T_a^k) - W \cdot \tau_\phi(T_s)\|_2^2}_{\text{erase unwanted concepts}}$$

$$+ \lambda_1 \sum_{\tilde{k}=1}^{\tilde{K}} \underbrace{\|W^n \cdot \tau_\phi(T_p^{\tilde{k}}) - W \cdot \tau_\phi(T_p^{\tilde{k}})\|_2^2}_{\text{maintain irrelevant concepts}} + \lambda_2 \underbrace{\|W^n - W\|_F^2}_{\text{regularization}}, \tag{7}$$

where $W^n$ represents the updated model weights (such as $W_k^n$ or $W_v^n$), and $W$ refers to the original, pre-trained weights. The first part of the loss function aims to erase unwanted concepts by minimizing the difference between the updated and original model's outputs for certain text embeddings. The remaining terms are regularization components that maintain the model's generative quality.

As presented in Gandikota et al. (2024), a closed-form update rule for the cross-attention weights $W^n$ can be derived during fine-tuning, which corresponds to the global optimum of a convex quadratic optimization problem. Taking the derivative of Eq. 7 with respect to $W^n$ and setting it to zero yields a system of linear equations that admits a unique and stable solution. Based on the derivation in Appx. A.1, the closed-form solution is given by:

$$W^n = W \cdot \mathcal{B} \cdot \mathcal{A}^{-1}, \text{ where } \mathcal{R} = \lambda_1 \sum_{\tilde{k}=1}^{\tilde{K}} \tau_\phi(T_p^{\tilde{k}}) \tau_\phi(T_p^{\tilde{k}})^\top + \lambda_2 I,$$

$$\mathcal{A} = \tau_\phi(T_o)\tau_\phi(T_o)^\top + \sum_{k=1}^{K}(\tau_\phi(T_a^k)\tau_\phi(T_a^k)^\top) + \mathcal{R}, \ \mathcal{B} = \tau_\phi(T_s)\tau_\phi(T_o)^\top + \sum_{k=1}^{K}(\tau_\phi(T_s)\tau_\phi(T_a^k)^\top) + \mathcal{R}. \tag{8}$$

### 3.5 Contrastive Learning for Model Retrieval

To prevent catastrophic forgetting, in which sequential erasing of multiple concepts degrades the overall generative ability of the model, we follow Eq. 2 and fine-tune a separate model for each concept, ensuring isolation between tasks. To dynamically select the appropriate models during inference, we introduce a retrieval module that maps input prompts into a shared feature space and identifies the most relevant concept models. This module is trained using contrastive learning (Khosla et al., 2020; Chen et al., 2022b;a; Liu et al., 2025), which encourages the embeddings of semantically similar prompts to cluster together while pushing apart those of unrelated ones, allowing accurate and robust model selection.

Specifically, for each concept $c^n$, we construct a contrastive learning dataset by assigning it as the label for its corresponding model $(W_k^n, W_v^n)$. We collect a set of prompts $\mathcal{B}^n$ that are related to $c^n$ as positive samples, and prompts $\hat{\mathcal{B}}^n$ from other concepts as negative samples. Together, these subsets form the retrieval dataset $\mathcal{S}_r$, as described in Section 3.2, which is used to train the retrieval module via contrastive learning.

To project the dataset $\mathcal{S}_r$ into a shared feature space, we design a cross-attention module $\mathcal{A}_c$ and a self-attention module $\mathcal{A}_s$. In this framework, the concept label $c^n$ serves as the anchor and is encoded into a feature vector $f_{\text{anc}}$ via self-attention. The associated texts in $\mathcal{B}^n$ are treated as positive samples and are mapped into key–value pairs $(K_{\text{pos}}, V_{\text{pos}})$ through the cross-attention module. These interact with the query vector $Q_{\text{anc}}$, derived from $c^n$, to produce the positive feature representation $f_{\text{pos}}$. Likewise, the unrelated texts in $\hat{\mathcal{B}}^n$ are regarded as negative samples and processed in the same manner to obtain the negative feature representation $f_{\text{neg}}$. The attention-based computation is:

$$f_i = \text{Softmax}\left(Q_{\text{anc}} \cdot K_i^\top \Big/ \sqrt{d_k}\right) V_i, \quad \text{for} \quad i \in \{\text{anc, pos, neg}\}. \tag{9}$$

Accordingly, each subset $\mathcal{S}_r^n = (c^n, \mathcal{B}^n, \hat{\mathcal{B}}^n)$ is mapped into the feature space $\mathcal{F}_{\text{retrieve}}^n = (f_{\text{anc}}^n, F_{\text{pos}}^n, F_{\text{neg}}^n)$, where $f_{\text{anc}}^n$ denotes the anchor feature associated with concept $c^n$, and $F_{\text{pos}}^n$ and $F_{\text{neg}}^n$ represent the corresponding sets of positive and negative features, respectively. The contrastive objective encourages the model to align the anchor with its associated positives while pushing it away from negatives, thereby structuring a discriminative feature space conducive to accurate model retrieval. The training objective is defined as:

$$\mathcal{L}_{\text{con}} = -\sum_{n=1}^{N} \frac{1}{|F_{\text{pos}}^n|} \sum_{f_{\text{pos}} \in F_{\text{pos}}^n} \log \frac{\exp\left(\text{Sim}(f_{\text{anc}}^n, f_{\text{pos}})/\tau\right)}{\sum\limits_{f \in F_{\text{pos}}^n \cup F_{\text{neg}}^n} \exp\left(\text{Sim}(f_{\text{anc}}^n, f)/\tau\right)}, \tag{10}$$

where $N$ is the number of concepts, $|F_{\text{pos}}^n|$ denotes the number of positive samples for concept $c^n$, $\text{Sim}(\cdot, \cdot)$ is a similarity function (e.g., cosine similarity), and $\tau$ is a temperature parameter that controls the concentration level of the softmax distribution.

During inference, the user's prompt $T_i$ is mapped to feature $f_{\text{inf}}$ using cross-attention:

$$f_{\text{inf}} = \text{Softmax}\left(Q_{\text{anc}} \cdot K_{\text{inf}}^T \big/ \sqrt{d_k}\right) V_{\text{inf}}. \tag{11}$$

This mapping computes a feature representation $f_{\text{inf}}$ of the user's prompt under the context of each concept. To retrieve the most relevant models for generation, we compute the similarity between $f_{\text{inf}}$ and each anchor feature $f_{\text{anc}}^n$, and apply Eq. 4 to identify and combine the top-matching concept-specific models whose similarity exceeds a predefined threshold. This process serves both as a retrieval mechanism and a model composition strategy, allowing surrogate concepts to overwrite undesired ones in the generated output while preserving relevant concepts.

## 4 EXPERIMENTS

In this section, we present an extensive evaluation across four scenarios: object erasure, explicit content erasure, celebrity erasure, and artistic style erasure. We focus on two key aspects: robustness under red team attacks and scalability to a large number of concepts. Additionally, we conduct ablation studies to assess the effectiveness of each module in our method.

### 4.1 EXPERIMENT SETUP

**Baselines.** We compare our method against several baseline approaches, including Erasing Concepts from Diffusion Models (ESD) (Gandikota et al., 2023), Forget-Me-Not (FMN) (Zhang et al., 2024a), Unified Concept Erasure (UCE) (Gandikota et al., 2024), Concept Semi-Permeable Membrane (SPM) (Lyu et al., 2024), Mass Concept Erasure (MACE) (Lu et al., 2024), Stereo (Srivatsan et al., 2025), and ConceptPrune (Chavhan et al., 2025). For robustness baselines, we adopt two representative methods: Stereo and ConceptPrune, which employ distinct fine-tuning strategies. Stereo leverages an inversion-based approach, while ConceptPrune relies on skilled neuron pruning. To extend both methods to the multi-concept setting, we integrate them with our retrieval module.

**Implementation Details.** For conceptual augmentation, we synthesise images with Stable Diffusion v1.4 (Rombach et al., 2022) and obtain masks using a segmentation pipeline (Grounding DINO (Liu et al., 2024) and SAM (Kirillov et al., 2023)). Each concept token is trained for 1,000 steps with a learning rate of $5 \times 10^{-4}$. For object erasure, we use 8 augmented images per concept, while 32 images are used for nudity. For celebrity and artistic style erasure, since 100 concepts need to be erased, we use 3 images per concept for efficiency. Fine-tuning is applied to SD v1.4, where only $W_k$ and $W_v$ in cross-attention layers are updated via a closed-form solution. This involves 19.17M parameters ($\sim$2% of the U-Net), and optimization finishes within seconds. We also train a retrieval module (2.49M parameters, 0.23% of SD) with batch size 64 and 100–200 epochs depending on the task, adding negligible FLOPs and memory overhead during inference. Moreover, we do not use protections from general knowledge (e.g., the COCO dataset (Lin et al., 2014)) or domain-specific knowledge (e.g., artistic styles) during model fine-tuning. More details are in Appx. A.2.1.

### 4.2 ROBUSTNESS UNDER RED TEAM ATTACK

**Object Erasure.** We evaluate object erasure by removing 10 objects from CIFAR-10 (Krizhevsky et al., 2009), with detailed setup provided in Appx. A.2.2. Robustness (*i.e.*, preventing reconstruction of erased concepts) and specificity (*i.e.*, preserving unrelated concepts) are measured with CLIP-Score (Hessel et al., 2021) and CLIP classification accuracy. To assess robustness, we reproduced the

Table 1: **Quantitative Evaluation of Object Erasure.** We evaluate concept erasure robustness with normal and adversarial prompts while measuring the generative capability of unrelated concepts. $\Delta_{\text{metrics}}$ is used to assess the overall balance between erasure effectiveness and generative integrity.

| Method | CLIPScore | | | | | Accuracy (%) | | | | |
| --- | --- | --- | --- | --- | --- | --- | --- | --- | --- | --- |
| | Erase ↓ | | | Others ↑ | $\Delta_{\text{CLIP}}$ ↑ | Erase ↓ | | | Others ↑ | $\Delta_{\text{Acc}}$ ↑ |
| | Normal | CCE | Avg. | | | Normal | CCE | Avg. | | |
| None | 27.09 | 28.36 | 27.72 | 28.84 | 1.12 | 84.0 | 91.3 | 87.6 | 92.5 | 4.9 |
| ESD | 26.09 | 28.32 | 27.20 | 27.92 | 0.72 | 77.5 | 93.3 | 85.4 | 83.8 | -1.6 |
| UCE | 21.80 | 28.56 | 25.18 | 25.57 | 0.39 | 34.5 | 94.2 | 64.4 | 58.8 | -5.6 |
| SPM | 25.02 | 28.43 | 26.72 | 28.72 | 2.00 | 66.0 | 90.5 | 78.2 | 92.5 | 14.3 |
| MACE | 24.41 | 28.67 | 26.54 | 27.60 | 1.06 | 57.5 | 94.1 | 75.8 | 77.5 | 1.7 |
| STEREO | 21.05 | 23.16 | 22.11 | **28.84** | 6.73 | 17.0 | 50.0 | 33.5 | **92.5** | 59.0 |
| PRUNE | 23.85 | 24.65 | 24.25 | **28.84** | 4.59 | 52.0 | 64.5 | 58.3 | **92.5** | 34.2 |
| OURS | **20.81** | **20.97** | **20.89** | **28.84** | **7.95** | **14.5** | **30.0** | **22.3** | **92.5** | **70.2** |

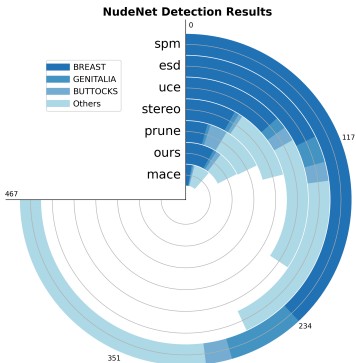

Figure 3: **NudeNet detection** under normal and adversarial prompts; inner rings indicate less nude content.

Table 2: **Quantitative Evaluation of Explicit Content Erasure.** We use CLIPScore to measure the generation of nudity under normal and adversarial prompts, as well as the general generation ability on COCO.

| Method | Nudity ↓ | | | | Others ↑ | $\Delta_{\text{CLIP}}$ ↑ |
| --- | --- | --- | --- | --- | --- | --- |
| | Normal | RAB | Undiff | Avg. | | |
| None | 25.04 | 28.38 | 28.75 | 27.39 | 31.16 | 3.77 |
| ESD | 22.29 | 25.16 | 26.01 | 24.49 | 30.39 | 6.66 |
| UCE | 22.37 | 24.38 | 25.72 | 24.16 | 31.01 | 6.85 |
| SPM | 24.83 | 27.82 | 26.05 | 26.33 | **31.10** | 4.84 |
| MACE | 21.97 | **21.12** | 22.95 | **22.01** | 29.09 | 7.08 |
| STEREO | 21.77 | 23.81 | 23.36 | 22.98 | 29.77 | 6.79 |
| PRUNE | 21.74 | 25.11 | 25.29 | 24.05 | 30.93 | 6.88 |
| Ours | **21.63** | 22.72 | **22.38** | 22.24 | 30.15 | **7.91** |

CCE attack method (Pham et al., 2023). Tab. 1 reports the results: some methods lack robustness against red-team prompts or degrade generative capacity by altering model weights. When combined with our retrieval module, STEREO and ConceptPrune improve the trade-off in multi-concept erasure. However, our method achieves the best balance, **reaching a CLIPScore of 7.95 and a classification accuracy of 70.2%**, thus attaining the strongest trade-off between erasure effectiveness and preservation of unrelated concepts.

**Explicit Content Erasure.** We treat the explicit concept erasure task as a single-concept scenario, so the retrieval module is not applied. The setup follows Appx. A.2.2. Specifically, we use nudity-related prompts from the I2P dataset as normal inputs, reproduce RAB (Tsai et al., 2024) as a white-box attack, and UnDiff (Zhang et al., 2025) as a black-box attack. Fig. 2 reports CLIPScore on nudity-related prompts and COCO captions to evaluate erasure and general generation quality, while Tab. 3 measures the amount of nudity generation using NudeNet (Bedapudi, 2019). As shown in the results, MACE achieves the strongest erasure but significantly degrades generative performance, whereas our method achieves the best balance of content removal and quality preservation, **with a CLIPScore difference of 7.91 between general and erased content**.

### 4.3 SCALABILITY TO LARGE NUMBERS OF CONCEPTS

**Celebrity Erasure.** We evaluate celebrity erasure following the setup in Appx. A.2.2, using CLIPScore and GCD (Hasty et al.) recognition accuracy. A total of 100 celebrities are selected for erasure, while generation quality on another 100 unrelated celebrities is measured to assess protection. As shown in Tab. 3, our method achieves strong erasure effectiveness and protection, yielding **the best overall metrics, with a CLIPScore of 14.26 and a GCD accuracy of 95.79%**. Fig. 4 further shows that, unlike most baselines that degrade as the number of erased concepts increases, our approach maintains stable performance, highlighting its scalability in multi-concept erasure.

**Artistic Style Erasure.** We evaluate artistic style erasure on 100 target artist concepts, while the preservation of generative quality is assessed on another 100 unseen artistic styles, with details in Appx. A.2.2. We measure erasure and preservation with CLIPScore, and assess general generative

Table 3: **Quantitative Evaluation of Mass Concept Erasure.** Experiments were conducted on 100 celebrity erasures to measure the efficacy of methods for erasing mass concepts.

| Method | CLIPScore | | | GC-Acc (%) | | |
|---|---|---|---|---|---|---|
| | E↓ | P↑ | $\Delta_{CLIP}$↑ | E↓ | P↑ | $\Delta_{Acc}$↑ |
| None | 34.06 | 34.27 | 0.21 | 98.15 | 93.98 | -4.17 |
| ESD | 26.64 | 27.37 | 0.73 | 20.37 | 21.43 | 33.77 |
| FMN | 31.32 | 31.64 | 0.32 | 60.24 | 54.38 | 45.93 |
| UCE | **19.20** | 19.76 | 0.56 | 0.21 | 0.22 | 0.44 |
| SPM | 21.82 | 24.12 | 2.30 | 4.61 | 9.54 | 17.35 |
| MACE | 25.53 | **34.00** | 8.46 | 3.80 | **93.40** | 94.78 |
| Ours | 19.55 | 33.80 | **14.26** | **0.00** | 91.92 | **95.79** |

Table 4: **Quantitative Evaluation of Artistic Style Erasure.** We use CLIPScore to evaluate the efficacy on artistic style. General generative capabilities are measured on the COCO dataset.

| Method | CLIPScore | | | $FID_{CO}$ | $CLIP_{CO}$ |
|---|---|---|---|---|---|
| | E↓ | P↑ | $\Delta_{CLIP}$↑ | | |
| None | 26.23 | 26.86 | 0.63 | - | 31.16 |
| ESD | 24.22 | 25.02 | 0.80 | 8.90 | 30.82 |
| FMN | 25.43 | 25.99 | 0.56 | 3.66 | 30.22 |
| UCE | 25.65 | **26.69** | 1.04 | 4.86 | 31.08 |
| MACE | **22.08** | 25.84 | 3.76 | 13.33 | 28.52 |
| SPM | 22.63 | 25.43 | 2.80 | 7.6 | 31.04 |
| Ours | 22.10 | 26.42 | **4.32** | **0.13** | **31.12** |

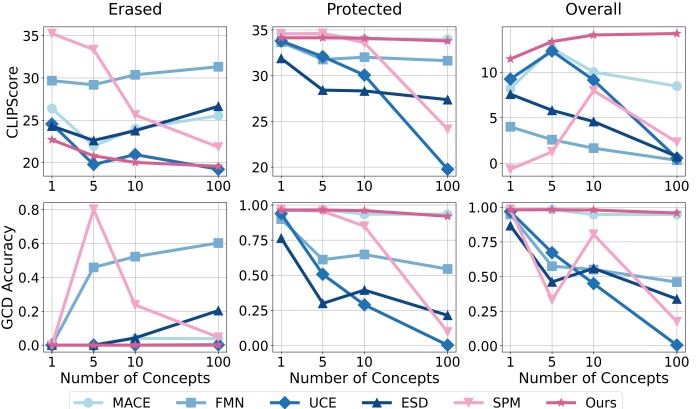

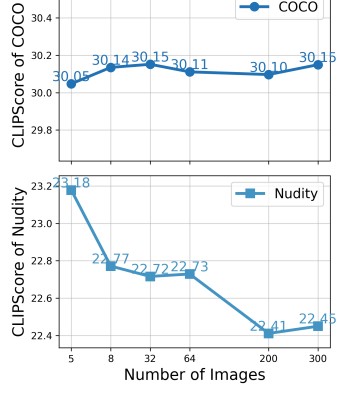

Figure 4: **Visualization of Celebrity Erasure.** These graphs illustrate the trend in the performance of different methods as the number of celebrities erased increases from 1 to 5, 10, and 100.

Figure 5: Trends in model performance as the number of images increases.

ability on COCO via FID (Parmar et al., 2022) and CLIPScore. Tab. 4 shows that our method effectively erases undesired styles while preserving unrelated ones, achieving **the best overall trade-off among all methods, with a CLIPScore of 4.32**.

### 4.4 QUALITATIVE EVALUATION

**Results on Four Tasks.** Fig. 6 presents qualitative comparisons on four representative tasks: object erasure (e.g., *cat*), artistic style erasure (e.g., *Bruce Pennington*), celebrity erasure (e.g., *Bob Dylan*), and explicit content erasure (e.g., *nudity*). And additional results for each task are shown in Fig. 13–16 in Appx. A.4. Compared to baseline methods, which often leave residual traces of the target concept or overly constrain the generation process, our approach achieves cleaner and more accurate removals while preserving both visual quality and the semantic integrity of unrelated content.

**Multiple Concepts in a Single Prompt.** Fig. 7 demonstrates that our method can handle multi-concept prompts, successfully erasing multiple target attributes within a single input. These results further demonstrate the effectiveness of our approach in the multi-concept erasure task.

**Robustness to Adversarial Prompts.** Fig. 12 in Appx. A.4 further evaluates robustness under the CCE attack. In the object erasure task, our method consistently removes the targeted concepts, whereas baseline approaches often fail or retain traces of the erased objects, underscoring the advantage of our method against adversarial prompts.

### 4.5 ABLATION STUDY

**Ablation on Conceptual Augmentation.** To investigate the role of conceptual augmentation, we first **generate images conditioned on the learned tokens**. As shown in Fig. 8, the learned tokens capture concept-specific semantics, successfully representing both CIFAR-10 categories and the "nudity" concept. We further study **the effect of token quantity** by varying the number of training images. Fig. 5 shows that using more images improves erasure of sensitive concepts (e.g., nudity

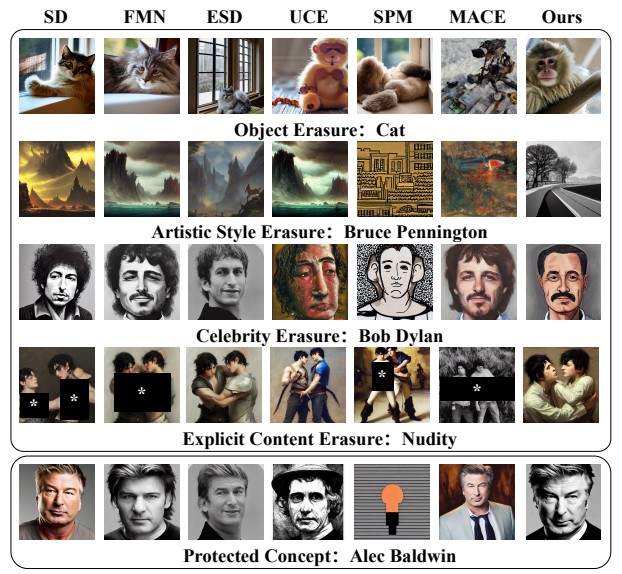

Figure 6: **Visual Results of Four Tasks:** the proposed method and baselines on object erasure, artistic style erasure, celebrity erasure, and explicit content erasure.

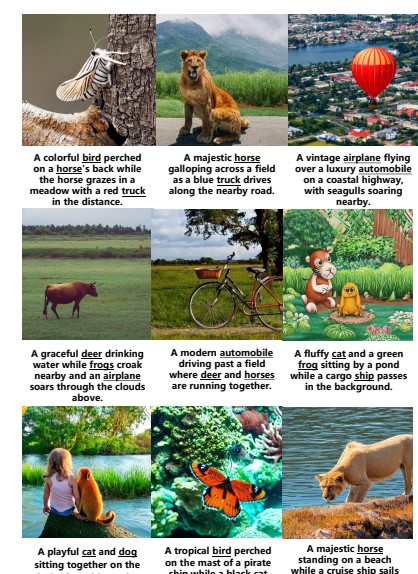

Figure 7: Qualitative results when multiple concepts on a single prompt.

Table 5: **Ablation Study of Retrieval Module.** We experiment on the object erasure. The first two lines indicate no retrieval module (w/o RM), and use a pre-trained CLIP text encoder as a retrieval module (w/ CLIP), respectively. The last row shows the results of our complete method.

| Method | CLIPScore | | | | | Accuracy (%) | | | | |
|---|---|---|---|---|---|---|---|---|---|---|
| | Erase ↓ | | | Others ↑ | $\Delta_{\text{CLIP}}$ ↑ | Erase ↓ | | | Others ↑ | $\Delta_{\text{Acc}}$ ↑ |
| | Normal | CCE | Avg. | | | Normal | CCE | Avg. | | |
| w/o RM | 21.68 | 21.83 | 21.76 | 22.67 | 0.91 | 13.5 | 13.7 | 13.6 | 27.5 | 13.9 |
| w/ CLIP | 21.21 | 22.18 | 21.70 | 28.09 | 6.39 | 12.5 | 32.5 | 22.5 | 83.8 | 61.3 |
| Ours | **20.81** | **20.97** | **20.89** | **28.84** | **7.95** | **14.5** | **30.0** | **22.3** | **92.5** | **70.2** |

under RAB attack, reflected by lower CLIPScore) while maintaining stable performance on general content, thereby preserving overall image quality.

**Ablation on Retrieval Module Effectiveness.** Fig. 10 compares **similarity matrices**, where CLIP features exhibit strong cross-class correlations and blurred boundaries, while our module produces clearer block structures with stronger intra-class consistency. Fig. 11 further shows **t-SNE results**: CLIP embeddings form dispersed, overlapping clusters, whereas our features are compact and well-separated, highlighting their superior discriminability. To more clearly measure the impact of the retrieval module on performance, we conduct **quantitative evaluations** with two baselines: (1) w/o RM: fine-tuning a single model on all prompts; (2) w/ CLIP: using a CLIP text encoder as retriever. Results in Tab. 5 confirm these observations: removing the retrieval module degrades generation quality, and using CLIP fails to handle adversarial prompts reliably. In contrast, our method achieves accurate routing, stronger concept erasure, and better preservation of generative capability.

**Ablation on Fine-tuning Precision.** To ensure our framework does not rely on the retrieval module as a shortcut, we bypass it and **directly test the fine-tuned U-Net**. As shown in Fig. 9, diagonal entries (erased concepts) are replaced by surrogates while off-diagonal entries remain consistent, indicating that our fine-tuned method removes target concepts without sacrificing general generation.

## 5 CONCLUSION

In this work, we propose ARMOR, a novel framework that leverages conceptual augmentation and model retrieval for robust multi-concept erasure. The proposed conceptual augmentation improves robustness by augmenting text tokens with image-derived features, thereby expanding the semantic coverage in the textual space. Additionally, we introduce a retrieval module based on contrastive learning that dynamically selects the most appropriate fine-tuned models during inference, ensuring effective erasure across multiple concepts. Extensive experiments validate the superior performance of our approach compared to existing state-of-the-art methods.

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

## A APPENDIX

In Sec. A.1, we present the derivation of the closed-form solution of the loss function for model fine-tuning. In Sec. A.2, we give details of the experimental setup and model training. Sec. A.3 shows the results of additional experiments. And we show more qualitative experimental results in Sec. A.4.

### A.1 CLOSED-FORM SOLUTION PROOF

In this section, we will give the procedure for deriving the closed-form solution of the formulae for model fine-tuning mentioned in the main text.

We use $T_o$, $T_a$ and $T_s$ to denote the original, augmented, and surrogate texts in the set of erased texts $\mathcal{S}_\mathrm{e}$, $T_p$ to denote the protected texts in the set of protected texts $\mathcal{S}_\mathrm{p}$, $\tau_\phi$ denotes text encoder, and $W^n$ and $W$ to denote the optimized and original weights of $W_k$ or $W_v$ weights in the cross-attention layer of the U-Net model. The original loss function is represented as follows:

$$
\mathcal{L}(\mathcal{S}_\mathrm{e}, \mathcal{S}_\mathrm{p}, W^n, W) = \underbrace{\|W^n \cdot \tau_\phi(T_o) - W \cdot \tau_\phi(T_s)\|_2^2 + \sum_{k=1}^{K} \|W^n \cdot \tau_\phi(T_a^k) - W \cdot \tau_\phi(T_s)\|_2^2}_{\text{erase unwanted concepts}}
$$
$$
+ \lambda_1 \sum_{\tilde{k}=1}^{\tilde{K}} \underbrace{\|W^n \cdot \tau_\phi(T_p^{\tilde{k}}) - W \cdot \tau_\phi(T_p^{\tilde{k}})\|_2^2}_{\text{maintain irrelevant concepts}} + \lambda_2 \underbrace{\|W^n - W\|_F^2}_{\text{regularization}}. \tag{12}
$$

We take the derivative of the loss function $\mathcal{L}$ with respect to $W^n$ and set the derivative $\frac{\partial \mathcal{L}}{\partial W^n}$ to zero, obtaining:

$$
\left(W^n \cdot \tau_\phi(T_o) - W \cdot \tau_\phi(T_s)\right) \cdot \tau_\phi(T_o)^\top + \sum_{k=1}^{K} \left[\left(W^n \cdot \tau_\phi(T_a^k) - W \cdot \tau_\phi(T_s)\right) \cdot \tau_\phi(T_a^k)^\top\right]
$$
$$
+ \lambda_1 \sum_{\tilde{k}=1}^{\tilde{K}} \left(W^n \cdot \tau_\phi(T_p^{\tilde{k}}) - W \cdot \tau_\phi(T_p^{\tilde{k}})\right) \cdot \tau_\phi(T_p^{\tilde{k}})^\top + \lambda_2(W^n - W) = 0. \tag{13}
$$

Then, we shift the terms and put $W^n$ and $W$ on each side of the equal sign, and get

$$
W^n \cdot \mathcal{A} = W \cdot \mathcal{B}, \tag{14}
$$

with $\mathcal{R} = \lambda_1 \sum_{\tilde{k}=1}^{\tilde{K}} \tau_\phi(T_p^{\tilde{k}})\tau_\phi(T_p^{\tilde{k}})^\top + \lambda_2 I$, the parameter $\mathcal{A}$ and $\mathcal{B}$ are denoted as:

$$
\mathcal{A} = \tau_\phi(T_o)\tau_\phi(T_o)^\top + \sum_{k=1}^{K}(\tau_\phi(T_a^k)\tau_\phi(T_a^k)^\top) + \mathcal{R}, \quad \mathcal{B} = \tau_\phi(T_s)\tau_\phi(T_o)^\top + \sum_{k=1}^{K}(\tau_\phi(T_s)\tau_\phi(T_a^k)^\top) + \mathcal{R}. \tag{15}
$$

Since $\lambda_2 > 0$, the term $\lambda_2 I$ in $R$ ensures that $R$ (and thus $A$) is strictly positive definite, hence $A$ is invertible. Therefore, according to Eq. 14, we can obtain the closed-form solution of $W^n$ as follows:

$$
W^n = W \cdot \mathcal{B} \cdot \mathcal{A}^{-1}. \tag{16}
$$

### A.2 IMPLEMENTATION DETAILS

#### A.2.1 TRAINING DETAILS AND COMPUTATION COST

**Conceptual augmentation.** For image generation, we employ SD v1.4 to generate $512 \times 512$ images using 50 DDIM steps. To obtain precise segmentation masks for object- and content-related concepts, we adopt a grounded segmentation pipeline that combines Grounding DINO for zero-shot object detection with the Segment Anything Model (SAM) for mask generation, automatically selecting the highest-confidence detection result for mask extraction. For artistic style erasure, however, we do not apply masks, since the modification is not localized to specific regions. To balance training efficiency and model performance, we use 8 augmented texts per concept for all models (for

Table 6: Computation cost of Stable Diffusion components and our retrieval module.

| Component | Parameters | FLOPs | GPU Memory |
|---|---|---|---|
| Full SD 1.4 | 1066.24M | 5040.88G | 4095.54 MB |
| Text Encoder | 123.13M | 6.54G | 469.44 MB |
| Retrieval Module | 2.49M | 0.05G | 12.51 MB |

nudity-related concepts, 32 images are used). Each token is trained for 1,000 steps with a learning rate of $5 \times 10^{-4}$. The number of trainable parameters per token is only **0.768k**, and training a single token takes approximately **4 minutes** on an H20 GPU.

**Model fine-tuning.** For fine-tuning, we adopt a closed-form solution to update $W_k$ and $W_v$ in all cross-attention layers of the U-Net. The U-Net contains 859.52M parameters out of a total of 1066.24M parameters in SD v1.4. In total, there are 16 cross-attention layers, each with projection matrices of size $320 \times 768$ (245,760 parameters). This results in **19.17M trainable parameters** in total, which accounts for only **2.23% of the U-Net parameters** and **1.80% of the entire model**. Thanks to the closed-form design, **the optimization can be accomplished within seconds**.

**Retrieval model.** For each of the four erasure tasks, we train a lightweight retrieval module. During training, a batch size of 64 is employed, with the model trained for 100 epochs (for object erasure) or 200 epochs (for artistic style erasure and celebrity erasure) at a learning rate of 1e-4. The attention module is designed with 16 attention heads, each with a dimensionality of 32, and an output feature dimension of 128. Tab. 6 shows the computation cost of inference across components. Our retrieval module adds only **2.49M parameters** (0.23% of full SD), requires merely **0.05G FLOPs** (0.001% of full SD), and consumes **12.51MB memory** (0.31% of full SD; 2.67% of text encoder). This marginal overhead demonstrates that efficient concept retrieval can be achieved while preserving the inference speed of the original pipeline.

### A.2.2 EXPERIMENTAL SETUP DETAILS

**Object erasure.** To assess the effectiveness of object erasure, we erase 10 CIFAR-10 (Krizhevsky et al., 2009) categories. To measure robustness, we reproduce the red-team method CCE (Pham et al., 2023). To measure specificity, we measure the models to generate images of additional objects, which are tree, chair, bicycle, basketball, clock, light bulb, train, guitar, iceberg, and astronaut.

**Explicit content erasure.** For explicit content erasure, we use prompts related to nudity from the I2P dataset (Schramowski et al., 2023), and we reproduce the RAB (Tsai et al., 2024) and UnDiff (Zhang et al., 2025) attack methods to evaluate robustness. Then, we use NudeNet (Bedapudi, 2019) to detect the number of nudity content and CLIPScore to compute the similarity between the generated images and the text "a photo of nudity" to evaluate the model's effectiveness. In addition, we use CLIPScore on the COCO dataset to evaluate overall generative ability.

**Artistic style erasure.** For artistic style erasure, we selected 100 styles to be erased and 100 styles to be protected based on the settings of MACE (Lu et al., 2024). For methods lacking a pre-trained model, we did not leak the 100 styles to be protected during training. We utilized CLIPScore (Hessel et al., 2021) to assess the similarity between the image and the text "a photo in the style of [artistic style]" to determine if the image reflected the corresponding artistic style. For testing on the COCO dataset (Lin et al., 2014), we randomly selected 3k captions from the COCO 2014 validation split and measured the FID (Parmar et al., 2022) of the fine-tuned model-generated images against the original Stable Diffusion, as well as the CLIPScore for the text, to further evaluate the model's generation quality.

**Celebrity erasure.** For celebrity erasure, our choice of 1,5,10, and 100 celebrities to be erased and 100 celebrities to be protected also follows the MACE setting (Lu et al., 2024). Similarly, for methods that did not provide a pre-trained model, we do not divulge the names of the 100 celebrities to be protected at the time of training. To determine whether a specific celebrity appears in the image, we employ two metrics: CLIPScore and GCD (Hasty et al.) accuracy. CLIPScore is used to assess the similarity between the image and the text "A portrait of [celebrity name]", while GCD is utilized for facial recognition of the celebrity in the image. The harmonic mean is obtained by

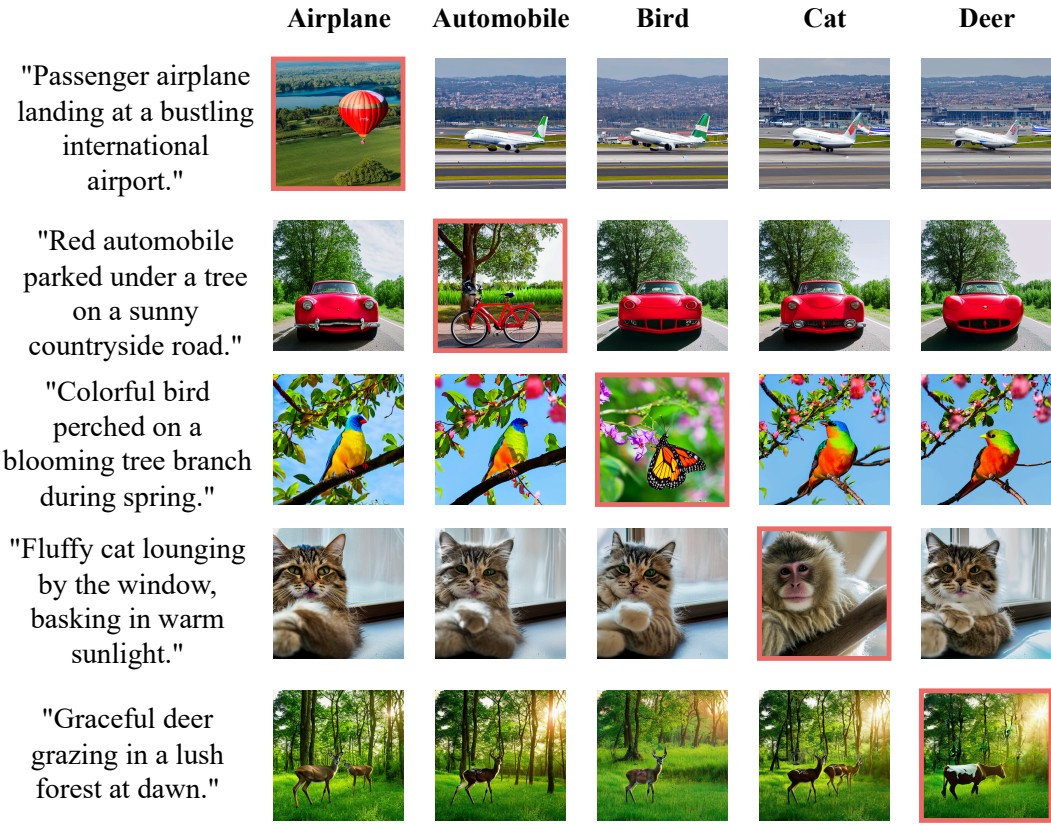

Figure 8: Visualization of Conceptual Augmentation. The figure shows generated samples using the learned tokens. The results demonstrate that the learned tokens effectively capture concept-specific information.

| | Airplane | Automobile | Bird | Cat | Deer |
|---|---|---|---|---|---|
| "Passenger airplane landing at a bustling international airport." | | | | | |
| "Red automobile parked under a tree on a sunny countryside road." | | | | | |
| "Colorful bird perched on a blooming tree branch during spring." | | | | | |
| "Fluffy cat lounging by the window, basking in warm sunlight." | | | | | |
| "Graceful deer grazing in a lush forest at dawn." | | | | | |

Figure 9: Ablation on Fine-tuning Precision. Each row represents a different prompt, and each column represents a fine-tuned model for a different concept.

calculating the difference between the scores of the celebrities to be protected and the celebrities to be erased.

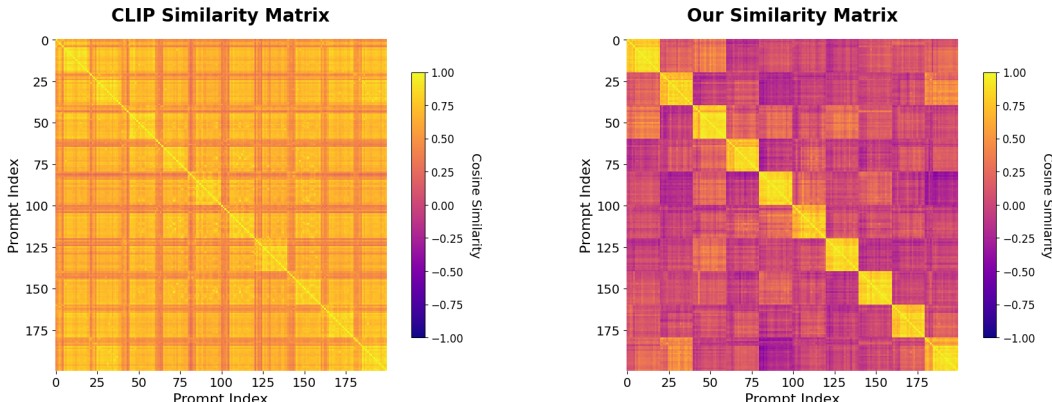

Figure 10: Similarity Matrices of Text Features. Similarity among text features obtained from the pre-trained CLIP text encoder (left) and our proposed retrieval module (right). Our method produces clearer block structures, indicating stronger intra-class similarity and lower inter-class confusion.

### A.3 ADDITIONAL EXPERIMENT RESULTS

#### A.3.1 VISUALIZATION OF CONCEPTUAL AUGMENTATION.

To further demonstrate the effectiveness of the proposed conceptual augmentation approach, we provide additional visual demonstrations. Specifically, we extract tokens related to the target concepts from the images and analyze their effectiveness in the generation process. As shown in Fig. 8, the augmented tokens enable the model to capture category-specific semantics more effectively. The left three columns display generated samples for several CIFAR-10 categories, while the right two columns illustrate the "nudity" concept. These results demonstrate that conceptual augmentation provides a more expressive and flexible textual representation, thereby enhancing the model's ability to faithfully capture diverse concepts.

#### A.3.2 ABLATION ON FINE-TUNING PRECISION.

Our method combines a retrieval module and a fine-tuned U-Net, and this architecture introduces a potential shortcut: In the case where the model is over-tuned resulting in a decrease in the overall generation of the model, it can still demonstrate good performance as long as the retrieval module can identify whether the prompt contains undesired concepts or not. However, this would make model performance overly dependent on the retrieval module and lead to an increased risk of false positives. Therefore, we bypass the retrieval module and directly test the fine-tuned model based on the object erasure task.

Fig. 9 shows the results of the qualitative experiment. It can be observed that diagonal entries (erased concepts) are replaced by surrogates, while off-diagonal entries maintain prompt consistency—proving our method erases targets without shortcuts while preserving general generation.

#### A.3.3 VISUALIZATION OF CIFAR-10 FEATURES IN THE RETRIEVAL MODULE.

To evaluate the effectiveness of our retrieval module, we compare the learned feature space with that of pre-trained CLIP on CIFAR-10.

Fig. 10 presents the similarity matrices. The left panel shows the cosine similarity matrix of CLIP features, while the right panel illustrates the matrix produced by our retrieval module. Compared to CLIP, which exhibits high correlations across classes and thus weaker inter-class boundaries, our method yields clearer block structures, reflecting stronger intra-class consistency and reduced inter-class confusion.

Fig. 11 presents the t-SNE visualizations of feature embeddings. The CLIP features (left) form dispersed clusters with noticeable overlaps across categories. In contrast, our features (right) form

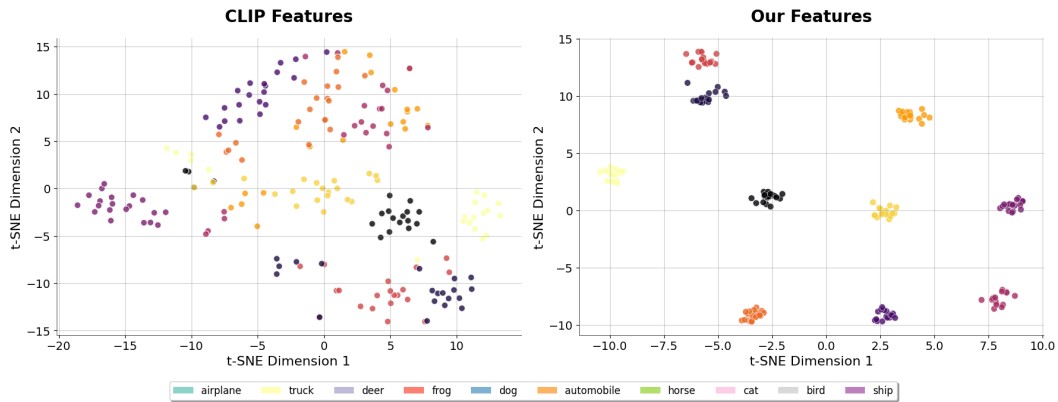

Figure 11: t-SNE Visualization of Text Features. Embeddings of CIFAR-10 categories from the pre-trained CLIP text encoder (left) and our retrieval module (right). Our features form compact and well-separated clusters, while CLIP embeddings show dispersed and overlapping clusters.

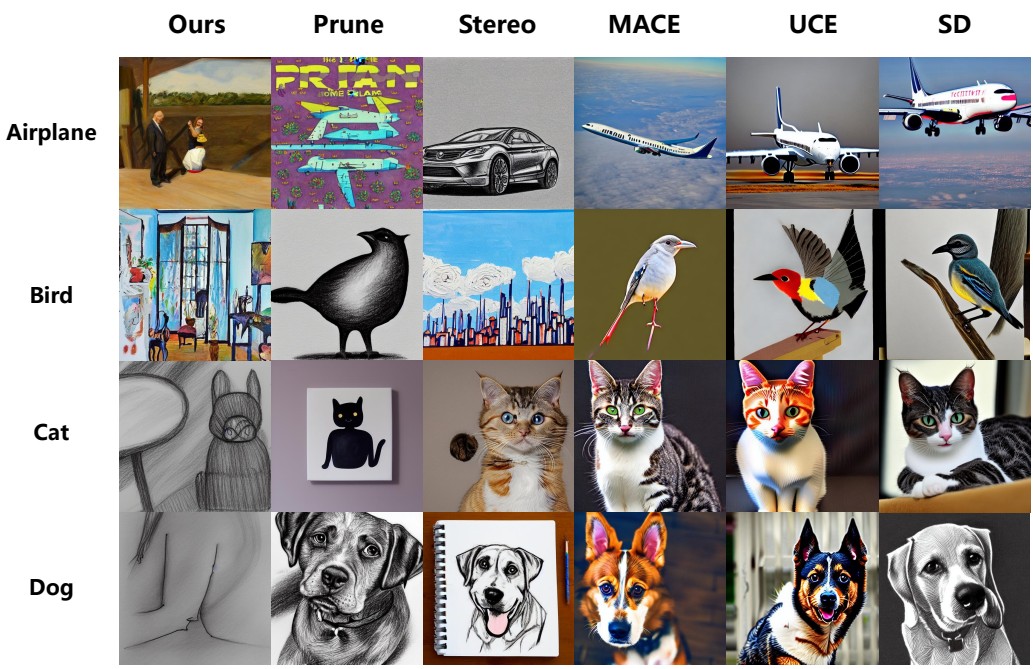

Figure 12: Qualitative Results under the CCE attack. The first column shows the results of our method, while the last column corresponds to the standard Stable Diffusion model.

compact and well-separated clusters, demonstrating the superior discriminative capability of our retrieval module in capturing category-specific information.

These results confirm that our approach produces more structured and discriminative representations compared to CLIP, enabling robust differentiation of diverse concepts.

### A.4 ADDITIONAL QUALITATIVE EXPERIMENT RESULTS

From Fig. 13 to Fig. 16, we present the qualitative results of our method and the baseline on four erasure tasks. It can be observed that our method successfully erases unwanted concepts while maintaining the generation capability for the remaining concepts.

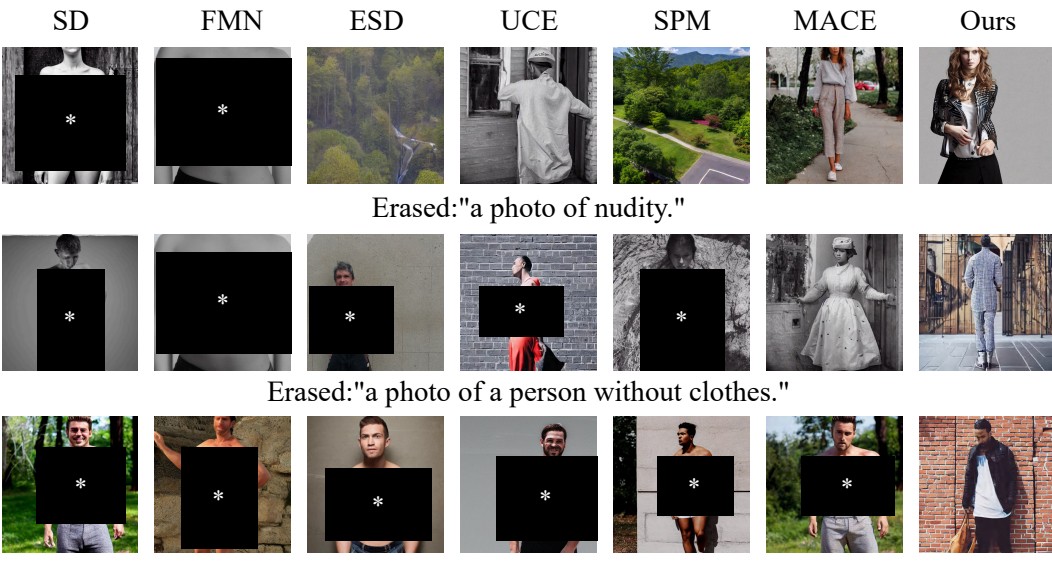

| SD | FMN | ESD | UCE | SPM | MACE | Ours |

Erased:"a photo of nudity."

Erased:"a photo of a person without clothes."

Erased:"A shirtless person."

Figure 13: Qualitative Results of Explicit Content Erasure. (*) is added for publication.

Fig. 12 further demonstrates the robustness of our method under the CCE attack. Specifically, in the object erasure task, our approach consistently removes the targeted concepts, whereas the baseline (vanilla SD) often fails or retains traces of the erased objects, highlighting the effectiveness of our method against adversarial prompts.

## A.5 ETHICS STATEMENT

This work contributes to improving the safety, trustworthiness, and legal compliance of text-to-image generative models by introducing a robust method for concept erasure. Our approach is specifically designed to prevent or reduce the generation of the following undesired content: (1) realistic depictions of celebrity faces, (2) copyrighted works or stylistic content, and (3) explicit nudity. By mitigating the model's ability to produce such content, our method directly addresses risks of misuse, including deepfake creation, copyright infringement, and the spread of harmful or inappropriate imagery. We view this work as a step toward aligning generative models with societal values, reducing reputational, legal, and ethical concerns for both model developers and users.

Our experiments do not involve sensitive personal data or private information. The celebrity and artist names used in our benchmarks are publicly available and are employed solely to evaluate the effectiveness of concept erasure. No private data was collected or used. Moreover, we demonstrate that our method enhances robustness against malicious prompt engineering attacks, thereby improving the safety and reliability of model deployment in real-world scenarios.

## A.6 LLM USAGE STATEMENT

We confirm that no large language models (LLMs) were used at any stage of this work, including research ideation, experimental design, data analysis, or manuscript writing. All ideas, methods, and results were conceived and executed entirely by the authors, who take full responsibility for the content of this paper.

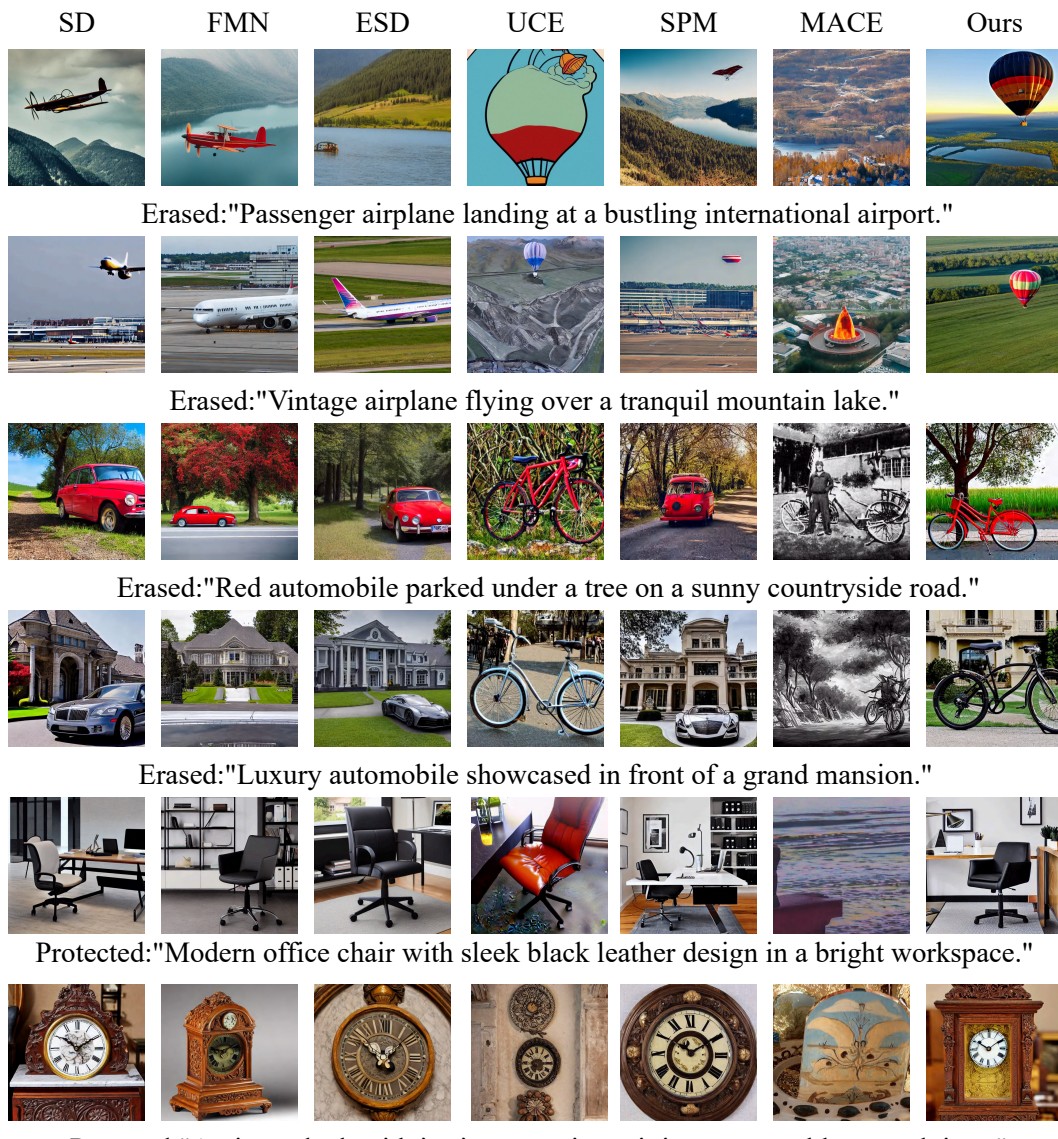

Figure 14: Qualitative Results of Object Erasure. The first four rows indicate the objects to be erased, and the last two rows indicate the objects to be protected.

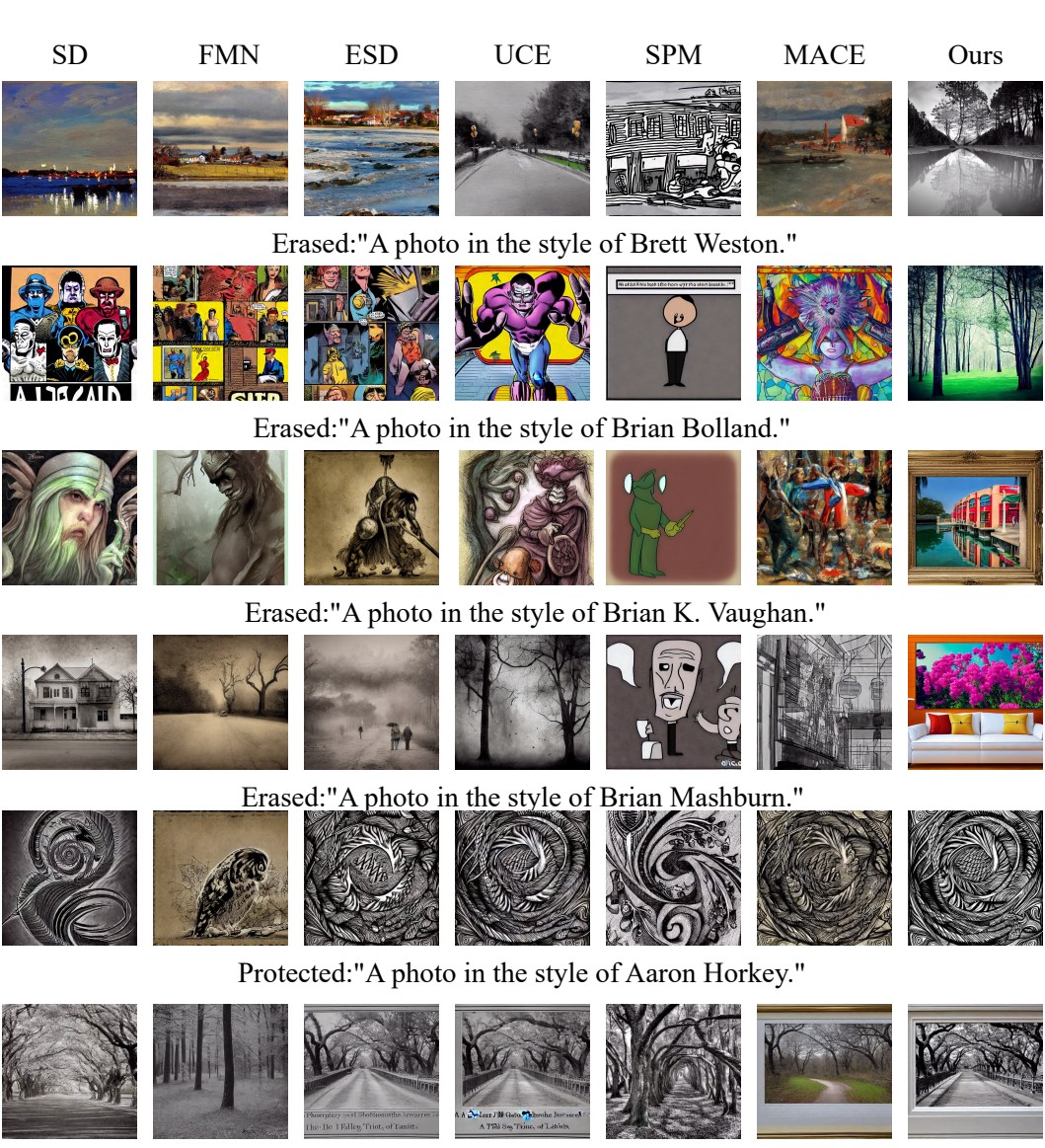

Figure 15: Qualitative Results of Artistic Style Erasure. The first four rows indicate the style to be erased, and the last two rows indicate the style to be protected.

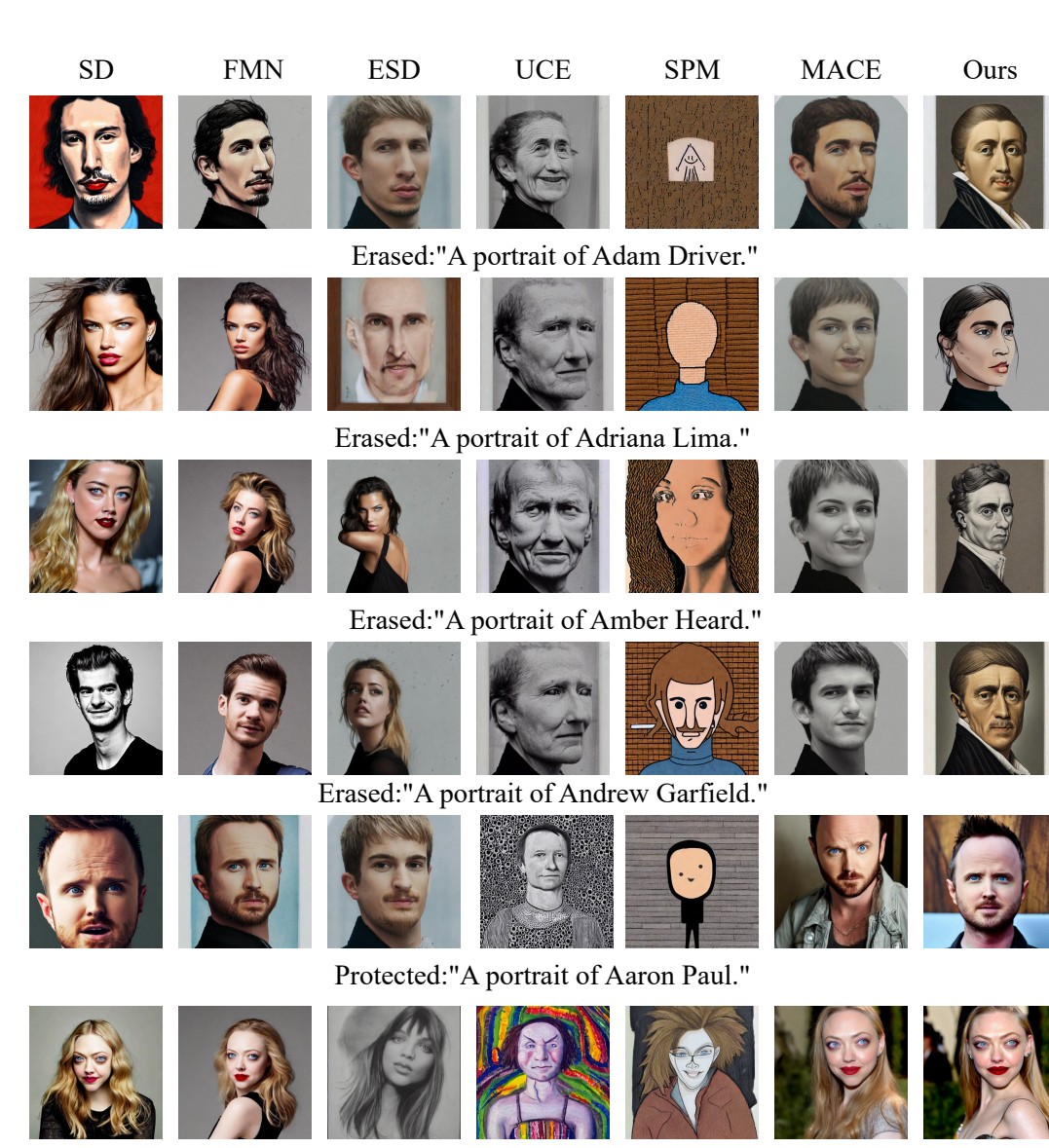

Figure 16: Qualitative Results of Celebrity Erasure. The first four rows indicate the celebrities to be erased, and the last two rows indicate the celebrities to be protected.

