# OpenReview forum: "ARMOR: Conceptual Augmentation for Robust Multi-Concept Erasure in Stable Diffusion via Model Retrieval"
_ICLR.cc/2026/Conference — ICLR 2026 Conference Withdrawn Submission_

### Official Review · Reviewer_dDsR · 2025-10-24

**Soundness:** 2
**Presentation:** 3
**Contribution:** 1
**Rating:** 2
**Confidence:** 5

**Summary:**

This paper proposes **ARMOR**, a framework for robust and scalable multi-concept erasure in Stable Diffusion. The authors identify two key challenges in existing concept erasure methods: vulnerability to adversarial or synonymous prompts (robustness) and severe degradation when erasing multiple concepts (multi-concept forgetting). ARMOR addresses these by combining **conceptual augmentation**, which back-optimizes text tokens against concept-related images to enrich textual representations, and **model retrieval**, which fine-tunes separate key/value cross-attention layers per concept and uses contrastive learning to select the most relevant “eraser” models during inference. Experiments across four benchmarks—object, explicit content, celebrity, and artistic style erasure—show that ARMOR achieves superior erasure robustness and general image quality, outperforming recent baselines such as MACE, UCE, and SPM by large margins in both quantitative and qualitative results.

**Strengths:**

1. The paper is well-organized, with detailed mathematical formulation (including a closed-form update for fine-tuning) and a clear pipeline diagram that illustrates the conceptual and retrieval stages.
2. The paper identifies two critical and underexplored challenges in concept erasure: robustness to adversarial prompts and scalability to large numbers of erased concepts, both of which are critical in concept erasure.

**Weaknesses:**

1. The proposed ARMOR framework is somewhat incremental and lacks substantial new insights. In Section 3.3 (*Concept-Augmented Dataset Construction*), the preprocessing pipeline largely follows prior work—*STEREO* employs textual inversion for adversarial training, and *MACE* uses SAM for object masking. In Section 3.4 (*Per-Concept Model Fine-Tuning*), the closed-form optimization is essentially a variant of *UCE* without introducing new analytical terms. As a result, the main technical novelty lies in Section 3.5 (*Contrastive Learning for Model Retrieval*). However, as elaborated below, this module is conceptually misaligned with the paper’s original motivation.
2. The authors argue that concept erasure is more effective and harder to bypass than post-filtering approaches, which I agree with—post-processing filters are impractical in real-world concept erasure applications. In black-box settings, simple rule-based or neural filters can already screen undesired content at input/output, while in white-box settings these filters (e.g., the *Stable Diffusion* safety checker) can be easily disabled. From this perspective, the proposed model retrieval module essentially acts as a post-filter mechanism. Regarding the two motivations stated in the paper:
   (a) For **robustness**, this module can be trivially bypassed in white-box scenarios, rendering it ineffective against adversarial attacks such as *CCE* or *UnlearnDiff*.
   (b) For **multi-concept erasure**, the improved performance mainly stems from decomposing the multi-concept task into multiple single-concept submodels, which is unsurprising. In fact, a simpler keyword-based matching strategy (e.g., matching celebrity names) might yield comparable results. Overall, the retrieval module contradicts the authors’ motivation and weakens the contribution.
3. In Table 2, using CLIPScore to evaluate *nudity* erasure may not be reliable. Most prior works use **accuracy (by NudeNet)**as the main metric for NSFW removal, while CLIPScore is rarely adopted for this purpose. Furthermore, CLIP itself is not well-trained on NSFW data, since such samples are typically filtered out during training, leading to inaccurate text-image alignment for explicit content. Although Figure 3 reports ASR results under “normal” and “adversarial” prompts, the distinction between the two cases is unclear. It would be better to follow standard benchmarks and report **ASR-based metrics** for a fair comparison.
4. In Figure 3, *MACE* achieves the best erasure performance, yet ARMOR is described as an incremental extension of it. It is unclear why ARMOR performs worse in robustness—does the use of textual inversion or other augmentations have a negative effect? In addition, *STEREO*’s reported results differ significantly from its original paper under the same benchmark. The authors should clarify what causes these discrepancies (e.g., different data splits or training hyperparameters).
5. In Table 3 and Figure 4, the performance of *UCE* deviates sharply from its reported results, despite *UCE* also being a multi-concept erasure method. The ΔAcc value (0.44) is much lower than expected, although its protection performance in Table 4 appears normal. It is unclear whether the authors aligned the experimental setup—for example, using 100 celebrities as the retain set for celebrity erasure and 100 styles for style erasure. Since *UCE* only provides the retain set for style erasure in its official release, the authors should double-check the experimental alignment to ensure fairness.

**Questions:**

See the weakness part.

---

### Official Review · Reviewer_MacP · 2025-10-29

**Soundness:** 1
**Presentation:** 2
**Contribution:** 1
**Rating:** 2
**Confidence:** 4

**Summary:**

This study is motivated by the need for robust and scalable concept erasure methods, i.e. methods to erasure certain concepts from text-to-image diffusion models. The main methodological clue of the presented ARMOR approach is the application of Textual Inversion (TI) to map a generated set of images back into the input embedding space as a way of augmenting the erasure target concept. It is claimed that this inversion-based concept augmentation improves the efficacy of the erasure. The erasure algorithm itself appears to be the closed-form approach from UCE so there is no contribution on the core algorithm but the presented augmentation is demonstrated to improve the robustness through the help of textual inversion, which in itself is quite similar to what STEREO (Srivatsan et al. CVPR'25) is doing but without the closed-form erasure process. When it comes to the multi-concept erasure scenario, they propose an inference-time retrieval mechanism that compares the input prompt to the erasure target of the already existing individual adapters, but their custom retrieval model is only minimally better than taking CLIP off-the-shelf. Overall, this work lacks novelty and timeliness as every individual component was already explored (closed-form erasure, textual inversion for target augmentation, retrieval of adapters) and their joint ARMOR approach is not achieving convincing results. It is only developed and tested for SD v1.4, while most existing research on new erasure methods focus already on newer architectures.  Most importantly, this work is motivated by "robustness" and "multi-concept erasure" but it is never actually demonstrated that ARMOR is capable of both at the same time.

**Strengths:**

- (S1) **Intuitive idea** of deriving a richer concept representation through augmentation by inversion.
- (S2) **Exploration of systematic approach** of using multiple adapters that are activated at inference-time based on the input prompt, following prior works like MACE or SPM.
- (S3) **Great range of erasure baselines and scenarios** that this study uses in its experiments to evaluate the proposed ARMOR method.

**Weaknesses:**

I find the following list of things to be major weaknesses:
- (W1) **Distinction to STEREO**: My current understanding: STEREO applies textual inversion during training to deeply collect adversarial examples that it can patch during the erasure, while ARMOR "mines" these augmented examples broadly before the erasure. STEREO uses an ESD-inspired non-linear erasure while ARMOR uses a linear UCE-based erasure. If that is true, then the main baseline is STEREO (which is missing in Figures 13 and 14, see W5). Generally, ARMOR does not seem to outperform STEREO; however, ARMOR has an advantage: it is likely cheaper or faster due to the internal closed-form erasure inherited from UCE. To answer whether ARMOR and STEREO are on par, it is important to test ARMOR against CCE. I would greatly appreciate some clarifying comments on this from the authors or other reviewers.
- (W2) **No results for reliably multi-concept erasure**: Table 3 suddenly does not show any robustness metrics, even though the motivation of this work is robust and scalable concept erasure. There are better approaches for robustness and simpler (such as using CLIP for retrieval) or already existing other methods for scalability (such as MACE).
- (W3) **Lack of clarity on the key methodological contribution**: Section 3.4 seems to be largely explaining the methodology of prior work (such as UCE). The proposed approach to applying contrastive learning for a retrieval mechanism proves to be less successful compared to simply using pre-existing CLIP as a retriever. The relevant contribution is thus the inversion-based augmentation, which I think needs to be a more prominent focus of this work with more, specific experiments for this particular part of the contribution. The reader wants to understand what role this augmentation actually plays, how many augmentations one needs, does this number differ between scenarios, and how it compares to more naive augmentation approaches like just rephrasing the prompt, adding noise to the embedding, or using synonyms or translations.
- (W4) **Unfair multi-concept erasure comparison to baselines**: Fine-tuning a separate model for each target and then using a retrieval module makes the comparison unfair to many of the other methods that do in-weight unlearning without adding additional multiple additional adapters/parameters. When the method relies on multiple of these adapters to be ready at inference time, then the LoRA adapters cannot be merged back into the model, which fundamentally changes the model and is thus only applicable to API-based black-box scenarios where users do not have system or model access. For example, MACE merges all the target-specific adapters into a single one at the end.
- (W4) **Overall results are not convincing**: The results overall only suggest a slight superiority of ARMOR when it comes to the CLIPScore metric. However, even this advantage is far from clear, and CLIPScore is generally not a very informative or sensitive metric.

And here the minor weaknesses:
- (W5) **Lack of consistency in results**: No consistent comparison to baselines. Figures 13 and 14, for example, miss STEREO. Table 2 misses the most challenging metric: CCE robustness.
- (W6) **No results for SD v2, SD v3, or beyond**, while the field is slowly moving to newer models as SD v1 pales w.r.t. to the image quality and prompt adherence in comparison to those newer models.
- (W7) **Figure 7 lacks original samples** before the erasure with the same prompts.

**Questions:**

- (Q1) Table 1: It is a bit strange that STEREO, PRUNE, and ARMOR all have the same values for "Others". I would appreciate a small comment on that.
- (Q2) Wrong highlighting in Table 5! The lowest "Erase" accuracies are not achieved by "Ours", right?

---

### Official Review · Reviewer_W2oZ · 2025-11-03

**Soundness:** 3
**Presentation:** 2
**Contribution:** 3
**Rating:** 6
**Confidence:** 4

**Summary:**

The paper presents a new framework ARMOR to improve the reliability and scalability of concept erasure in text-to-image diffusion models. The method addresses two key issues: the limited robustness of existing approaches against adversarial or synonymous prompts, and the severe quality degradation that occurs when multiple concepts are erased. ARMOR introduces Conceptual Augmentation, which transfers visual features into the text domain to enrich semantic coverage and strengthen robustness, together with Model Retrieval, which fine-tunes the cross-attention key and value projections for each concept and employs a contrastive retrieval module to select the appropriate erasure parameters during inference. This design allows the model to suppress specific undesired concepts while preserving general image generation quality. Extensive experiments demonstrate that ARMOR achieves superior performance compared with state-of-the-art baselines, effectively resists red-team attacks, and delivers over 10 percent improvements in CLIPScore across multiple erasure benchmarks.

**Strengths:**

1. The proposed ARMOR framework combines conceptual augmentation and model retrieval in an elegant way. Conceptual augmentation enriches textual representations with visual information, while model retrieval dynamically selects fine-tuned submodels for different concepts, reducing interference and catastrophic forgetting. The closed-form fine-tuning approach is also efficient and theoretically sound.
2. The paper precisely identifies two major limitations of existing concept erasure methods, namely lack of robustness to adversarial or synonymous prompts and poor scalability when erasing multiple concepts. This motivation is well grounded and practically important for safe deployment of text-to-image models.
3. The paper provides extensive experiments across multiple concept types such as objects, styles, nudity, and celebrities. Results consistently show improved robustness, better image quality, and superior performance under adversarial settings compared to strong baselines.

**Weaknesses:**

1. The main quantitative metrics are CLIPScore and accuracy on erasure detection, which focus on semantic similarity but not on perceptual fidelity or human satisfaction. Metrics such as FID could provide a fuller picture.
2. The paper does not provide quantitative measurements of training time, inference latency, or memory cost, especially as the number of erased concepts grows. This omission weakens the claim of scalability.
3. Architectural generalization is untested, as all results are reported on Stable Diffusion v1.4 without validation on SDXL or other diffusion backbones.
4. Ablations are incomplete, focusing on the retrieval module and a fine-tuning check, but lacking sensitivity to top-K  thresholds, regularization strengths $\lambda$, and the number of augmented tokens.

**Questions:**

1. How is the number of learned tokens per concept chosen in practice, and how sensitive is performance to this choice?
Other questions are consistent with weaknesses. If the author can solve the above problems, I will consider improving my score.

---

### Official Review · Reviewer_tM2s · 2025-11-06

**Soundness:** 3
**Presentation:** 3
**Contribution:** 2
**Rating:** 4
**Confidence:** 4

**Summary:**

This paper presents ARMOR, a framework for robust and scalable multi, concept erasure in diffusion models. The approach introduces three main components: (1) conceptual augmentation that expands textual representations using image, derived tokens via textual inversion, (2) lightweight per, concept fine, tuning of cross, attention key/value matrices through a closed, form update, and (3) a retrieval, based composition module that dynamically selects and blends relevant per, concept erasers at inference. Experiments are conducted on object, explicit, content, celebrity, and artistic, style erasure tasks, showing improved results compared to several existing methods (ESD, UCE, MACE, etc.).

**Strengths:**

1) Addresses a highly relevant challenge: scalable and robust concept erasure in diffusion models.
2) Well, structured and coherent framework combining conceptual token expansion, modular fine, tuning, and retrieval, based composition.
3) Demonstrates consistent improvements across multiple erasure categories and robustness evaluations.
4) Lightweight and modular design makes it efficient and easily extendable to large concept sets.
5) Clear motivation and strong practical relevance to safety and controllability in text, to, image generation.

**Weaknesses:**

Weaknesses
1) Missing direct comparison to the most relevant recent baseline (Receler).
The paper cites Receler but does not include experimental results against it. As Receler also focuses on reliable concept erasure with modular lightweight updates and retrieval, based activation, its absence prevents a fair and complete evaluation of ARMOR’s effectiveness.

2) Lack of clear evidence supporting the claimed novelty.
Several components closely parallel prior works: textual, inversion, based concept expansion (Kumari et al., ICLR 2023; Gandikota et al., ICCV 2023), selective cross, attention fine, tuning (ESD, ICCV 2023), and retrieval, based modular erasure (MACE, CVPR 2024; Receler, ECCV 2024). The paper would benefit from more explicit differentiation and justification of its unique contribution.

3) Limited discussion of challenging or failure cases.
While results are strong, the paper primarily highlights successful examples. A more balanced analysis of difficult multi, concept or visually overlapping scenarios would better establish reliability.

**Questions:**

Questions

1) How does ARMOR quantitatively compare to Receler under the same setup and datasets?

2) How are multiple per, concept Wₖ/Wᵥ matrices combined during inference without causing interference or instability?

3) Does the retrieval module ever produce false activations, and if so, how sensitive is performance to this?

---

### Note · Authors · 2025-11-19

I have read and agree with the venue's withdrawal policy on behalf of myself and my co-authors.